# Effect of In-Situ H Doping on the Electrical Properties of In$_2$O$_3$ Thin-Film Transistors

Peixuan Hu [1,2], Zhixiang Gao [2], Lu Yang [2], Wanfa Li [2], Xiaohan Liu [1,2], Ting Li [1,2], Yujia Qian [1,2], Lingyan Liang [2,*], Yufang Hu [1]  and Hongtao Cao [2,*]

1   State Key Laboratory Base of Novel Functional Materials and Preparation Science, School of Materials Science and Chemical Engineering, Ningbo University, Ningbo 315211, China; hupeixuan@nimte.ac.cn (P.H.); liuxiaohan@nimte.ac.cn (X.L.); liting@nimte.ac.cn (T.L.); qianyujia@nimte.ac.cn (Y.Q.); huyufang@nbu.edu.cn (Y.H.)
2   Laboratory of Advanced Nano Materials and Devices, Ningbo Institute of Materials Technology and Engineering, Chinese Academy of Sciences, Ningbo 315201, China; zxgao@mail.ustc.edu.cn (Z.G.); yjhylxyh814915@163.com (L.Y.); lwf078@mail.ustc.edu.cn (W.L.)
*   Correspondence: lly@nimte.ac.cn (L.L.); h_cao@nimte.ac.cn (H.C.)

**Abstract:** In this article, this research demonstrates the influence of in-situ introduction of H$_2$ into the working gas on the physical properties of post-annealed In$_2$O$_3$ thin films and the performance of associated devices. A gradual increase in the H$_2$ ratio leads to improved film quality, as indicated by spectroscopic ellipsometry, X-ray photoelectron spectroscopy, and atomic force microscope analyses showing a reduction in defect states such as band-tail states and $V_O$ in the film, and a smoother surface morphology with the root mean square roughness approximately 0.446 nm. Furthermore, this hydrogen doping effect results in a distinct shift in the device's threshold voltage toward the positive direction, and an improvement in the field-effect mobility and subthreshold swing. Consequently, a high-performance In$_2$O$_3$:H TFT is developed, exhibiting a field-effect mobility of 47.8 cm$^2$/Vs, threshold voltage of $-4.1$ V and subthreshold swing of 0.25 V/dec. These findings highlight the potential of in-situ H doping as a promising approach to regulate In$_2$O$_3$-based TFTs.

**Keywords:** thin-film transistors; metal oxide semiconductors; hydrogen; mobility; flat-panel displays

## 1. Introduction

Since the report by Nomura et al. on thin-film transistors (TFTs) using InGaO$_3$(ZnO)$_5$, oxide semiconductors (OSs) have been extensively researched due to their promising properties, such as wide band gap, excellent electrical properties, simple preparation, and good compatibility with Si-based manufacture. OSs have emerged as exceptional candidates for TFTs in pixel switching/driving applications for flat-panel displays (FPDs) [1]. However, while driving ultra-high-resolution FPDs at high frame rates, the field-effect mobility ($\mu_{FE}$) of oxide TFTs needs to be improved further [2]. In addition, other properties such as threshold voltage ($V_{th}$) and TFT stability are also critical factors that should be considered simultaneously [3]. Hence, it is essential to explore new OSs or fabrication techniques to improve the performance of OS TFTs.

Prototype OS materials used in TFT devices are indium oxide (In$_2$O$_3$), tin oxide (SnO$_2$), and zinc oxide (ZnO). Among them, In$_2$O$_3$-based OSs are considered as the most promising materials to achieve high mobility. Firstly, the conduction band bottom of In$_2$O$_3$ is composed of the 5s orbitals of the indium ion with a large orbital radius and a high overlap, which provides a highway for electron transportation [4,5]. Secondly, theoretical calculations show that In$_2$O$_3$ has a small and isotropic effective mass of electrons (0.22 $m_0$) [6]. Finally, some theories suggested that a higher In content results in a larger average effective coordination number of M–O bonds and more corner-, edge-, and face-shared M–O polyhedrons, which enhances overlapping of electron wavefunctions [3].

However, the excessively high carrier concentration (generally larger than $10^{18}$ cm$^{-3}$) of In$_2$O$_3$ makes it challenging to adjust electrical properties [4,7]. In order to regulate the electrical performance of In$_2$O$_3$-based TFTs, researchers have been studying extensively the effect of doping, which is considered to be the most promising method due to its effectiveness and simplicity.

Dopants can be categorized into cation and anion elements. In the case of cation doping, dopants with elements such as Ti, Hf, Zr, Bi, and W that have high oxygen bond dissociation energies, suppressed the concentration of excess oxygen vacancies ($V_O$), and then reduced the electron concentration in the films and improved the stability of the device [8–12]. Doping with lanthanide elements, such as Pr and Tb, shortened the relaxation time of photo-generated electrons and then enhanced the device's stability [13,14]. However, it should be noted that those cation dopants tend to have a somewhat degrading effect on the $\mu_{FE}$ of TFTs. On the other hand, anionic dopants like N, F, and H were also introduced to tune the film and the device's performance. It was found that nitrogen doping could enhance the device's stability by suppressing $V_O$ and other trap states [15,16], while fluoride could passivate $V_O$, suppress the formation of free carriers, and then improve stability [17–19]. And there are also a few reports on hydrogen doping. Wardenga et al. prepared H-doped In$_2$O$_3$ thin films (In$_2$O$_3$:H) by RF magnetron sputtering and found that hydrogen passivation of grain boundaries was the main reason for the high mobility observed in In$_2$O$_3$:H films [20]. Magari et al. conducted a study on anion doping with H and found that it plays a crucial role in increasing grain size and decreasing subgap defects, leading to an enhanced $\mu_{FE}$ [21]. In summary, it mainly contributed the $\mu_{FE}$ improvement to H-induced grain growth. Therefore, it is essential to further investigate H doping in In$_2$O$_3$, to supply more information on its effect on the film and device properties, such as the film microstructure, device $\mu_{FE}$, device threshold voltage ($V_{th}$), and others.

In this work, In$_2$O$_3$ and In$_2$O$_3$:H films were deposited by magnetron sputtering and their physical properties were investigated as a function of the H$_2$ ratio in the working gas. The results show that increasing the introduction of H$_2$ induces no grain growth, widens the band gap, makes the absorption edge steeper, and increases the percentage of M-O bonds. These variations correspond to a clear reduction in the electron concentration in In$_2$O$_3$. As a result, the TFT $V_{th}$ obviously shifts to the positive direction and the $\mu_{FE}$ is also improved with increasing H$_2$ ratio, showing that H doping is a promising way to regulate In$_2$O$_3$-based TFTs.

## 2. Experimental Methods

Heavily p-type doped Si wafers and 100 nm-thick SiO$_2$ were utilized as the gate electrode and insulator, respectively. The 25 nm In$_2$O$_3$ and In$_2$O$_3$:H channels were deposited using a 2-inch diameter In$_2$O$_3$ ceramic target (99.99% purity) through RF magnetron sputtering in a mixture of Ar, O$_2$, and H$_2$ gases at room temperature (RT). The O$_2$ and H$_2$ gas flow ratios are denoted as R[O$_2$] = O$_2$/(Ar + O$_2$ + H$_2$) and R[H$_2$] = H$_2$/(Ar + O$_2$ + H$_2$), respectively. For the In$_2$O$_3$ and In$_2$O$_3$:H film, R[O$_2$] was constant at 2.5%, R[H$_2$] was set to 0, 3, and 5%. The films were annealed at 350 °C for 1 h in air, and then a 100 nm-thick Al source/drain (S/D) layer was deposited through the direct current (DC) magnetron sputtering method using pure Ar gas at an input power of 65 W. The In$_2$O$_3$ and Al films were patterned using two different shadow masks. The width/length (W/L) was 800 μm/400 μm. Finally, the fabricated TFTs were annealed at 250 °C for 1 h in air. The optical absorption coefficient of the films was obtained through spectroscopic ellipsometry analysis (SE, M-2000 DI, J. A. Woollam, Lincoln, NE, USA), and the films' structural changes and crystallinity were evaluated via X-ray diffraction (XRD, D8 ADVANCE DAVINCI, Bruker, Karlsruhe, DE, USA) and transmission electron microscopy (TEM, Talos F200X, Thermo Fisher, Waltham, MA, USA). Surface morphology of the films was observed using an atomic force microscope (AFM, Dimension ICON, Bruker, Billerica, MA, USA). Hall effect testing was employed to determine various electrical properties of the thin films (8404-CRX-6.5K, Lake Shore, Woburn, MA, USA), whereas X-ray photoelectron spectroscopy (XPS, AXIS ULTRA DLD, Shimadzu, Kyoto, Japan) was utilized to investigate

internal properties and bonding character. A Keithley 4200SCS semiconductor parameter analyzer was employed to determine the electrical characteristics of the TFT devices.

### 3. Results and Discussion

Figure 1a displays the XRD patterns in $\theta$-$2\theta$ scans of 50 nm-thick unannealed $In_2O_3$ films with varying R[$H_2$]. The XRD spectra of as-deposited $In_2O_3$ films presented low diffraction intensity and rather broad background, indicating that the as-deposited films produced through magnetron sputtering are amorphous. After annealing in air at 350 °C for 1 h, all the films turn to the polycrystalline state; as observed in Figure 1b, there are four clear characteristic peaks which correspond to the (222), (400), (440), and (622) crystal planes [22]. These observed diffraction lines agree well with the cubic bixbyite indium oxide structure (JCPDS06-0416) and *Ia*-3 space group (Number: 206). Calculated with Scherrer's formula on the reflection (222), the grain sizes of the $In_2O_3$ films with 0%, 3%, and 5% hydrogen content are determined to be 50.6 nm, 48.2 nm, and 49.5 nm, respectively. Interestingly, as R[$H_2$] increases, the intensity of the (222) peak gradually lowers. These results indicate that the introduction of H in $In_2O_3$ does not promote grain growth, while it even more or less suppresses the crystallinity [23]. It is worth mentioning that these observations diverge from existing reports on H-doped $In_2O_3$ [21], which may be attributed to the discrepancy in the phases of the as-deposited $In_2O_3$ films: polycrystalline in the literature and amorphous in this work.

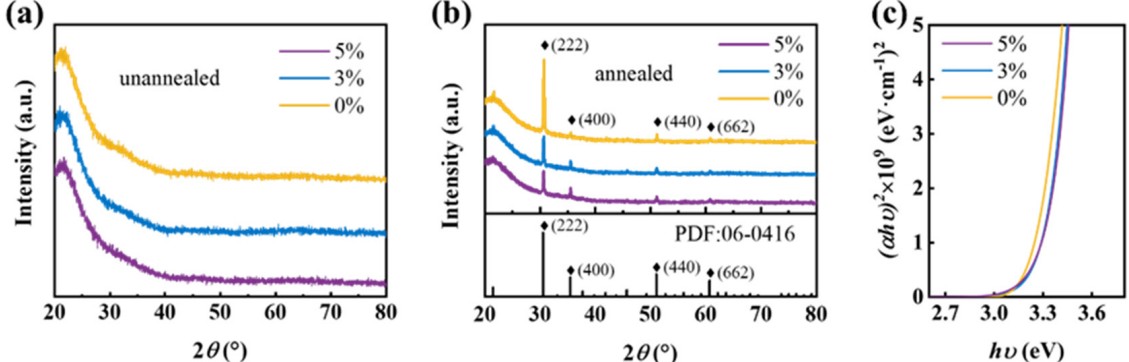

**Figure 1.** XRD patterns of the $In_2O_3$ and $In_2O_3$:H films deposited at different R[$H_2$] and at a constant R[$O_2$] value of 2.5% (**a**) before and (**b**) after annealing at 350 °C in air. Inset: (bottom) standard XRD spectra of $In_2O_3$. (**c**) Plots of $(\alpha h\nu)^2$ versus $h\nu$ of the annealed $In_2O_3$ and $In_2O_3$:H films.

Figure 1c presents the optical band gap ($E_g$) of the annealed films, which is estimated by the Tacu method ($\alpha h\nu \sim (h\nu - E_g)^n$) [24]. Since $In_2O_3$ has a direct band gap [25], then $n = \frac{1}{2}$ and thus $h\nu$ is estimated by extrapolating the linear $(\alpha h\nu)^2$ edge, where $h\nu$ represents the incident photon energy and $\alpha$ is the absorption coefficient obtained by spectroscopic ellipsometry analysis. The optical band gaps of $In_2O_3$ thin films are mostly reported in the range of 2.9–3.8 eV [7,23,25–28]. The calculated $E_g$ values of the $In_2O_3$:H films are very similar: about 3.28 eV, which is slightly larger than that of the $In_2O_3$ film (3.23 eV). And it can be observed that the absorption edges of the $In_2O_3$:H films are clearly steeper than that of the $In_2O_3$ film. These imply that the addition of hydrogen may be able to reduce the band-tail defects in $In_2O_3$, although it decreases the crystallinity of the $In_2O_3$ film.

In order to verify the crystallinity of the film, four representative samples were selected as undoped $In_2O_3$ and $In_2O_3$:H films with the highest doping concentration (R[$H_2$] = 5%) in the as-deposited and annealed states, respectively. The cross-sectional HR-TEM images of the four representative samples are presented in Figure 2a–d. For the as-deposited samples, the films exhibit an amorphous structure, consistent with the Fast Fourier Transformed results (inserts in Figure 2a,c), revealing diffused hollow rings indicative of the amorphous phase. Upon annealing, distinct grain boundaries and diffraction stripes were observed in Figure 2b,d. The domain-spacing (d-spacing) of the two labeled spherical particles in

the (222) crystal plane is about 2.91 and 2.84 Å, respectively, which is consistent with the standard cubic bixbyite-type structure of $In_2O_3$ (JCPDS06-0416). These results are in line with the results from the XRD test.

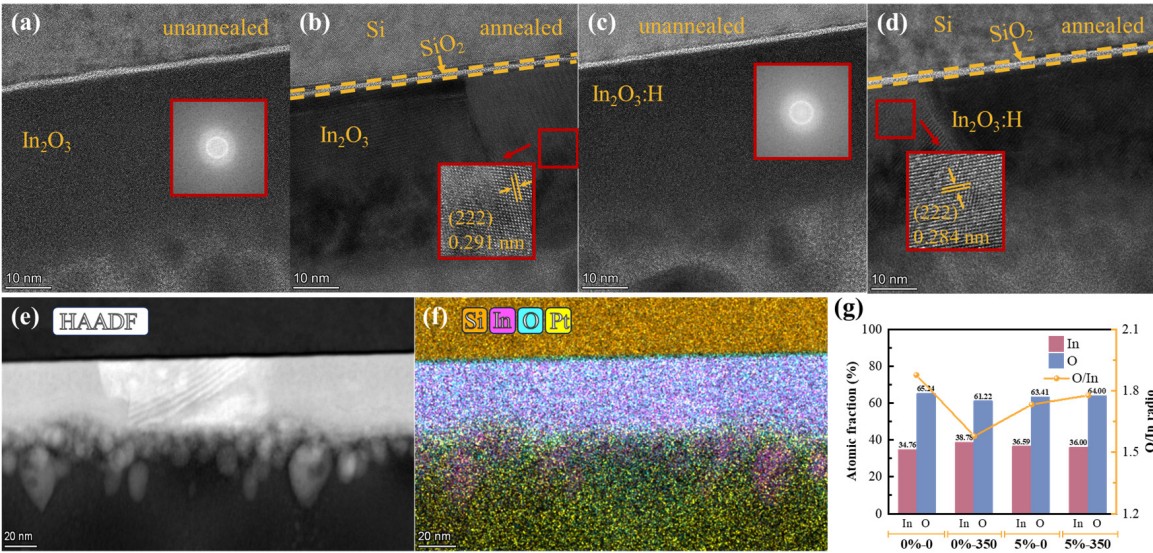

**Figure 2.** Cross-sectional HR-TEM images (**a**) 0%-0, (**b**) 0%-350, (**c**) 5%-0, (**d**) 5%-350. The insets to (**a**,**c**) show the Fast Fourier Transformed images of the as-deposited films. (**e**) HAADF image and (**f**) EDS mapping results of the annealed $In_2O_3$:H films at R[$H_2$] = 5%. (**g**) Atomic fractions of In, O, and the O/In radios in as-deposited and annealed $In_2O_3$ and $In_2O_3$:H films obtained from EDS analysis.

Figure 2e exhibits the high-angle angular dark-field (HAADF) image of $In_2O_3$:H films (R[$H_2$] = 5%); the interface between the Si layer and $In_2O_3$ layer is distinct and sharp. The thickness of the $In_2O_3$ layer is measured to be approximately 47.8 nm. This result is in good agreement with the simulated thickness which obtains from the spectroscopic ellipsometry analysis. One may also observe some large particles at the Pt region, which makes the $In_2O_3$/Pt interface unclear. These particles can be observed on all the samples chosen for TEM testing. Figure S1a shows the HAADF image of the as-deposited $In_2O_3$:H films at R[$H_2$] = 5%, where the particles are larger and their details are clearer compared to Figure 2c. EDS element analysis, shown in Figure S1b, shows that the element distribution of these large particles is indium-rich and oxygen-poor, which is the opposite of the element content inside the film. The atomic fractions (%) of Pt, O, and In are 2.78%, 25.89%, and 71.33%, respectively. By combining the two figures, it can be clearly observed that those large particles are separated from the indium oxide film region, seemingly suspended on the surface of this sample. This is because there is a distinct boundary between the particles and the $In_2O_3$ region, and the boundary is dominated with Pt. Figure 2f illustrates the corresponding energy-dispersive spectrometer (EDS) mapping for Si, In, O, and Pt elements, which reveals the homogeneous distribution of all elements. The In and O elements in the representative sample film were analyzed and the O/In ratio was calculated, as shown in Figure 2g. The results indicate minimal variation in the content of In and O across all samples. The O/In ratios of the hydrogen-doped films exhibit minimal change before and after annealing, in stark contrast to the undoped films. For the undoped films, upon thermal annealing at 350 °C, both the O content and the O/In ratios decrease and the $V_O$ increase, implying that some O atoms escape from their lattice positions, which is a frequent reported phenomenon for metal oxide semiconductors [29–31]. Conversely, in the case of H-doped films, the O/In ratios remain statistically unchanged between the as-deposited and annealed states. It is suggested that the introduction of hydrogen during the sputtering process may be the underlying cause of this phenomenon, as it significantly enhances the films' ability to fix oxygen. It has been mentioned in the literature that as the temperature

increases, the unstable interfacial In-H-In defects release hydrogen, which in turn combines with oxygen to form stronger covalent bonds, such as In-OH [32,33]. Furthermore, the O/In ratios of the annealed $In_2O_3$ film are observed to be lower than that of the annealed $In_2O_3$:H film.

Figure 3 shows the topographic profiles obtained by AFM scanning on both the as-deposited and annealed films at different hydrogen concentrations. A high-pass filter was applied to remove the swelling component on the substrate. The root mean square (RMS) roughness calculated from the AFM image is also displayed in Figure 3. The RMS values of the as-deposited films with hydrogen concentration of 0%, 3%, and 5% are 0.467, 0.350, and 0.307 nm, respectively. Notably, the test results reveal a gradual reduction in roughness with the introduction of hydrogen during the deposition process. After thermal annealing at 350 °C, the images showed that the surface morphology of $In_2O_3$ and $In_2O_3$:H thin films had undergone significant alterations, aligning well with the findings derived from XRD analysis. During the annealing process, the grains gradually aggregate to form grain clusters, and obvious boundaries are formed around the clusters. And the surface morphology of the undoped $In_2O_3$ film after annealing bears resemblance to the texture of a Hami melon peel. Compared with the annealed $In_2O_3$ film, the boundary surrounding the crystal clusters in the $In_2O_3$:H films becomes shallower, and the surface is notably smoother.

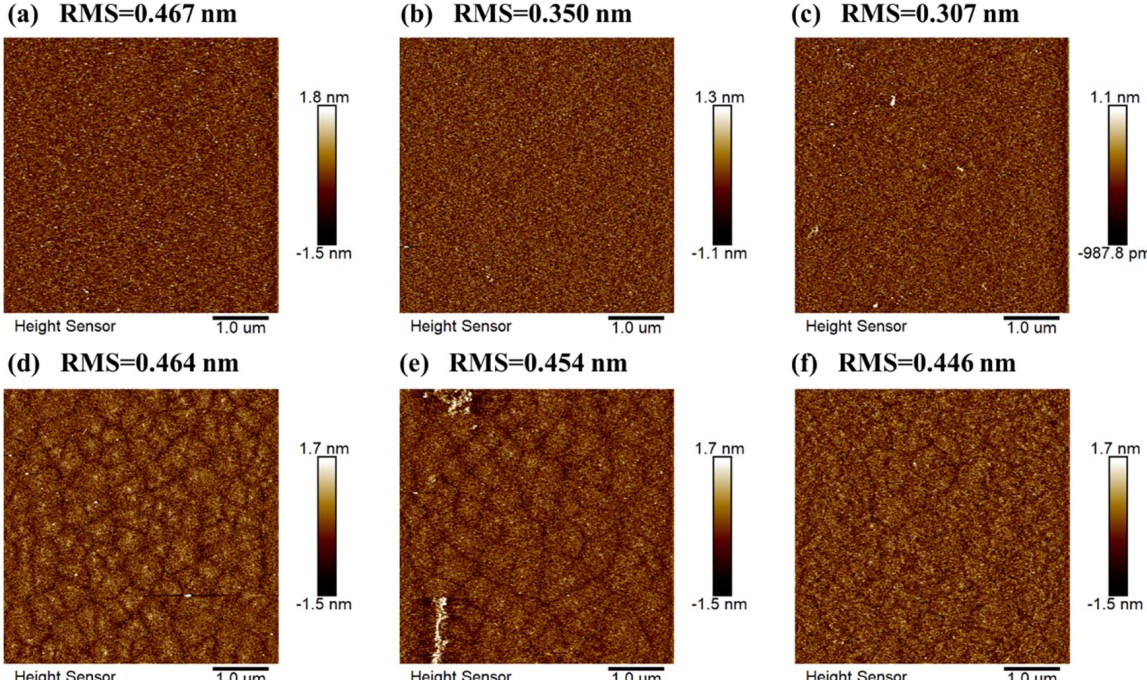

**Figure 3.** AFM images and root mean square roughness RMS (**a**) 0%-0, (**b**) 3%-0, (**c**) 5%-0, (**d**) 0%-350, (**e**) 3%-350, (**f**) 5%-350.

To further investigate the effect of hydrogen, XPS measurements were conducted. Considering that water, carbon, and other impurity molecules in the air will be adsorbed onto and even into the sample, a high dose of argon ions, specifically with an ion energy of 2 KeV and a current density of roughly 5 μA/cm², was applied to sputter away the top surface layer (about 20 nm), while the total thickness of the sample was 50 nm. Figure 4a–c shows the O 1s spectra for the inner layer of the annealed films. They are calibrated by taking the binding energy of the C 1s peak (284.8 eV) as a reference. Gaussian fitting is employed to deconvolute the O 1s peaks. The lowest binding energy Gaussian peak ($O_L$) at 529.8 eV originates from metal–oxygen bonds (M-O lattice); the peak at the middle binding energy 530.7 eV ($O_M$) is linked to $O^{2-}$ ions of oxygen defects, such as $V_O$; while the highest binding energy Gaussian peak ($O_H$) at 531.9 eV is usually associated with specific chemisorbed oxygen such as –$CO_3$, –OH, and $H_2O$ [34–36]. The area percentages of the

three O peaks were calculated and are summarized in Table 1. It was found that with increasing $R[H_2]$, the percentage of M-O bonds ($O_L$) increases while the recent part related to the defects and impurities ($O_M$ and $O_H$) decreases. This implies that in-situ H doping in the deposition of $In_2O_3$ film aids in improving film quality during the post-annealing process, especially by reducing $V_O$ or other impurity defects. It is also consistent with the EDS element content analysis that the $In_2O_3$:H film has less $V_O$ than the $In_2O_3$ film, which has higher oxygen content and a higher O/In ratio.

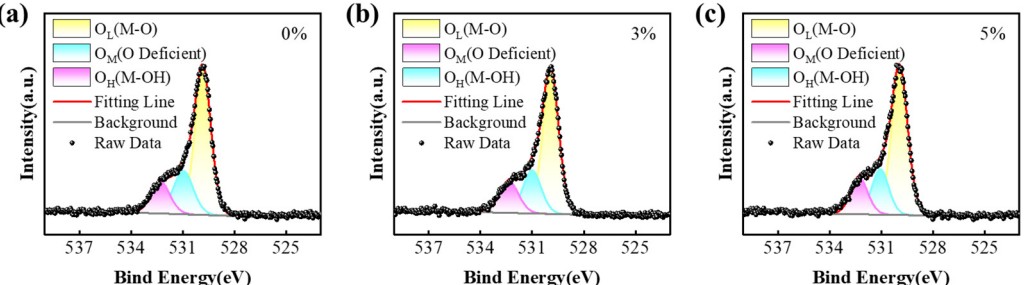

**Figure 4.** The XPS O 1s spectroscopic analysis of $In_2O_3$ etching films at a depth of 20 nm with different $R[H_2]$: (**a**) $R[H_2]$ = 0%; (**b**) $R[H_2]$ = 3%; (**c**) $R[H_2]$ = 5%.

**Table 1.** The percentage of O 1s peak area in XPS spectra and Hall effect testing results of the films.

| | In-O (%) | $V_O$ (%) | Impurity Molecules (%) | $\mu_H$ (cm²/Vs) | $N_e$ (1/cm³) | $\rho$ (Ω·cm) |
|---|---|---|---|---|---|---|
| 0% | 61.32 | 20.46 | 18.22 | 14.3 | $7.46 \times 10^{17}$ | 0.586 |
| 3% | 62.68 | 19.74 | 17.59 | 18.0 | $4.64 \times 10^{16}$ | 7.49 |
| 5% | 63.95 | 18.76 | 17.29 | 18.0 | $6.53 \times 10^{16}$ | 5.31 |

To investigate the electrical properties of the annealed $In_2O_3$ and $In_2O_3$:H films, Hall effect measurement was employed to determine the Carrier Hall mobility ($\mu_H$), electron concentration ($N_e$), and resistivity ($\rho$) of the thin films. The summarized data are presented in Table 1. The undoped film exhibits a low $\rho$ (0.586 Ω·cm) and a relatively high $N_e$ of $7.46 \times 10^{17}$/cm³. Upon the introduction of H, the $\rho$ increases by at least one order of magnitude, which mainly originates the decrease in the $N_e$, down to $4.64 \times 10^{16}$/cm³ at $R[H_2]$ = 3% and $6.53 \times 10^{16}$/cm³ at $R[H_2]$ = 5%. Additionally, the $\mu_H$ increased from 14.3 cm²/Vs for the $In_2O_3$ film to 18.0 cm²/Vs for the $In_2O_3$:H film. These variations imply that there are fewer donor defects available to provide free electrons in the $In_2O_3$:H films compared to the undoped $In_2O_3$ film. These findings also agree well with the results obtained from absorption spectra and XPS analysis that $V_O$ and band-tail states are clearly reduced by H doping, since these two kinds of defects are highly associated with donor states near the conduction band minimum of OSs. And the reduction in these defect states also brings an increase in mobility due to the reduction in carrier scattering centers. In addition, a smoother surface morphology also plays a role in improving the mobility of the device.

The fabricated $In_2O_3$-based TFTs' device structure is depicted in Figure 5a. The transfer characteristics of all annealed TFTs, displayed in Figure 5b, were measured at room temperature with a drain-source voltage ($V_{DS}$) of 10.1 V. $V_{th}$ was defined by gate voltage ($V_{GS}$) at a drain current ($I_D$) of 10 nA, and *SS* was extracted from $V_{GS}$, which required an increase in the $I_D$ average values from 0.1 nA to 10 nA. It is evident that as the $R[H_2]$ gradually increases, the $V_{th}$ of the TFTs significantly shifts toward the positive direction. Specifically, the $V_{th}$ changes from −15.0 V to −4.1 V. This observed shift in $V_{th}$ corresponds well with the decrease in $N_e$ of the films, as determined by the Hall measurement. What is more, the $\mu_{FE}$ of the device also exhibits an improvement trend with increasing $R[H_2]$. Herein, the $\mu_{FE}$ is derived from the transfer curves measured at $V_{DS}$ = 10.1 V, where the devices are working in the saturation region (as seen in the output

curves of Figure 5c). The $\mu_{FE}$ values increase from 37.8 cm$^2$/Vs to 47.8 cm$^2$/Vs, which is in good accordance with the corresponding trend observed for the $\mu_H$. It is worth noting that the $\mu_{FE}$ values are larger than the $\mu_H$ values due to the special carrier transportation mechanism corresponding to the percolation theory for OSs [37]. In TFTs, as the gate voltage increases, localized states become occupied with gate-induced carriers, causing the Fermi level ($E_F$) to shift toward the mobility edge ($E_m$). In such scenarios, the occupied localized states do not trap additional electrons. Particularly when the gate-induced carriers occupy a significant portion of the localized states and the electron density in the channel is sufficiently high to elevate the $E_F$ above $E_m$, electrons can move more freely, resulting in higher mobility compared to intrinsic films with lower $N_e$. As a result, the electrons can move relatively freely, leading to a higher mobility compared to the intrinsic films with lower $N_e$. Figure 5c shows the output characteristics of In$_2$O$_3$ (R[H$_2$] = 0%), In$_2$O$_3$:H (R[H$_2$] = 3%), and In$_2$O$_3$:H (R[H$_2$] = 5%) devices. The $I_D$ increases linearly at the region of low source-drain voltage ($V_{DS}$), indicating that an Ohmic contact formed between the S/D electrodes and channel layer. Clear pinch-off and saturation characteristics are evident in the saturation region of the output curve, indicating effective control of the transistor channel by the gate and source/drain. In$_2$O$_3$:H TFTs all exhibit high levels of output curve saturation current ($V_{DS}$ = 15 V, $I_D$ = 587 μA@$V_{GS}$ = 15 V at R[H$_2$] = 3%, $I_D$ = 560 μA@$V_{GS}$ = 12 V at R[H$_2$] = 5%). The $I_D$ of In$_2$O$_3$ TFT without introducing H$_2$ is decreased to 361 μA ($V_{GS}$ = 1 V, $V_{DS}$ = 15 V). It shows that with the addition of hydrogen, the TFT exhibits excellent current amplification performance.

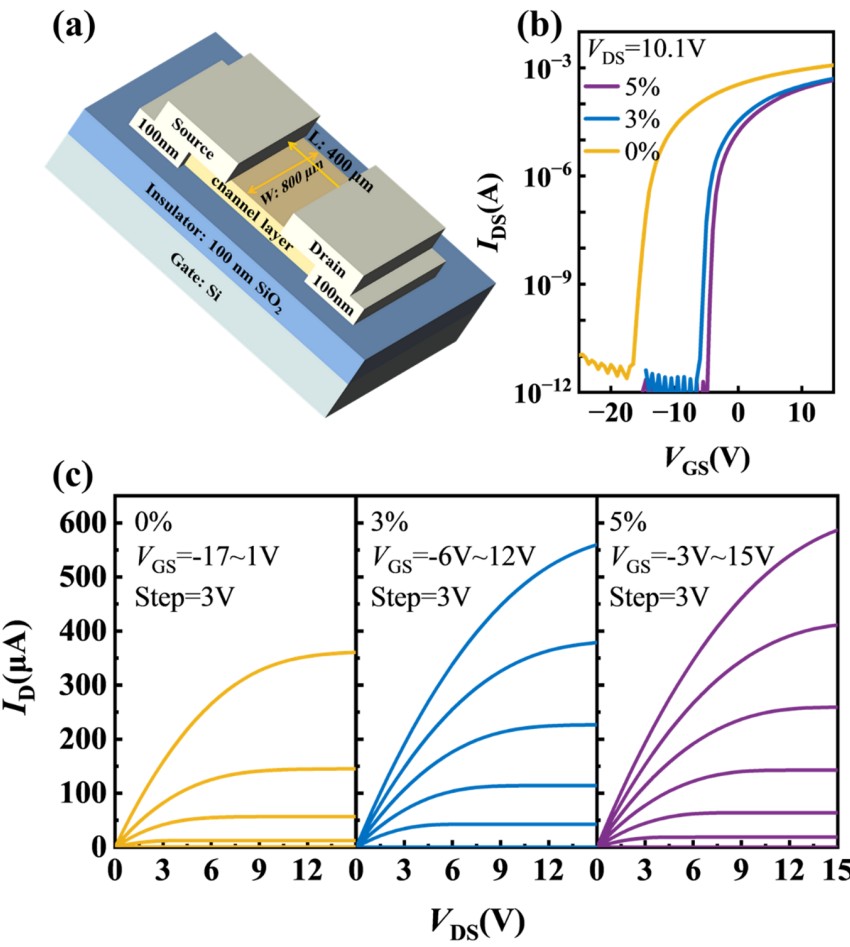

**Figure 5.** (**a**) Schematic view of a TFT device. (**b**) Typical transfer characteristics ($V_{DS}$ = 10.1 V) for all devices with In$_2$O$_3$ and In$_2$O$_3$:H channels deposited at various R[H$_2$] values. (**c**) Typical output curves of the TFTs.

In addition, as depicted in Figure 5b and detailed in Table 2, the off-state current ($I_{off}$) of the device subjected to the in-situ hydrogen doping process is significantly lower compared to that of the undoped device. As a result, the on–off current ratio ($I_{on}/I_{off}$) of the device shows a notable improvement, increasing specifically from $2.83 \times 10^7$ (R[H$_2$] = 0%) to $7.20 \times 10^8$ (R[H$_2$] = 5%), where the $I_{on}$ refers to the current value at $V_{GS}$ = 15 V on the transfer curve. These improvements can sensitively reflect changes in the channel defect states. It is worth noting that there is a distinct improvement in subthreshold swing (*SS*) from 0.52 V/dec to 0.25 V/dec after in-situ H doping. The *SS* is generally considered as an indicator of the sum of the defect states in oxide semiconductors and semiconductor/dielectric interface trap states. In this context, we introduce the concept of an effective maximum areal density of states, denoted as $N_{eff}$, which is calculated using the following formula [38]:

$$N_{eff} = \left( \frac{SS\log(\mathrm{e})}{kT/q} - 1 \right) \frac{C_{ox}}{q} \qquad (1)$$

where *k* represents the Boltzmann constant, *T* is the temperature, *q* is the electron charge, and $C_{ox}$ denotes gate oxide capacitance per unit area. The $N_{eff}$ were calculated to be $1.82 \times 10^{12}$, $9.46 \times 10^{11}$, and $8.76 \times 10^{11}$ cm$^{-2}$/eV for In$_2$O$_3$ TFTs deposited at R[H$_2$] = 5%, 3%, and 0%, respectively. These results indicate that an increase in R[H$_2$] leads to a reduction in the $N_{eff}$ of the device. Since the dielectrics and fabrication conditions used are the same just with H doping, it is believed that the semiconductor/dielectric interface trap states are very similar for the three devices, and the decrease in the $N_{eff}$ should correspond to the reduction in the defect states in oxide semiconductors: this is more strong proof that the film quality of In$_2$O$_3$ can be improved via in-situ H doping in as-deposited films.

**Table 2.** The main electrical parameters and derived effective defect density of states for TFT devices.

|  | $V_{th}$ (V) | $\mu_{FE}$ (cm$^2$/Vs) | $I_{on}/I_{off}$ | *SS* (V/dec) | $N_{eff}$ (cm$^{-2}$/eV) |
|---|---|---|---|---|---|
| 0%-350 | −15.0 | 37.8 | $2.83 \times 10^7$ | 0.52 | $1.82 \times 10^{12}$ |
| 3%-350 | −5.2 | 45.6 | $5.94 \times 10^8$ | 0.27 | $9.46 \times 10^{11}$ |
| 5%-350 | −4.1 | 47.8 | $7.20 \times 10^8$ | 0.25 | $8.76 \times 10^{11}$ |

## 4. Conclusions

In summary, polycrystalline In$_2$O$_3$ thin films and transistor devices were fabricated by room temperature magnetron sputtering with the introduction of H$_2$ into the working gas following an air annealing. Compared to the undoped one, the H-doped thin films show fewer defects (including oxygen vacancies and band-tail states) and a smoother surface morphology with similar average grain sizes. And the electron concentration obtained by Hall effect testing in the In$_2$O$_3$:H films is approximately one order of magnitude lower than that in the undoped films, while the Hall mobility is enhanced. Regarding thin-film transistors, the threshold voltage gradually shifts toward the positive direction and the field-effect mobility increases with an increase in H$_2$ ratio, agreeing well with the variation trends for the thin films. And the improved subthreshold swing values indicate a reduction in defect states in In$_2$O$_3$:H films. The optimized In$_2$O$_3$:H TFT demonstrates markedly improved electrical performance, including a high field-effect mobility of 47.8 cm$^2$/Vs, a more positive threshold voltage, and a smaller subthreshold swing of 0.25 V/dec. These results supply a comprehensive investigation on the property evolutions for H-doped In$_2$O$_3$ thin films and their TFTs, which would provide an opportunity to boost their further exploration and application.

**Supplementary Materials:** The following supporting information can be downloaded at: https://www.mdpi.com/article/10.3390/electronics13081478/s1, Figure S1: (a) HAADF image and (b) EDS mapping results of the as-deposited In$_2$O$_3$:H films at R[H$_2$] = 5%.

**Author Contributions:** Conceptualization, L.L.; Methodology, L.L.; Software, P.H. and L.Y.; Formal analysis, P.H., Z.G. and W.L.; Investigation, Z.G. and X.L.; Data curation, T.L.; Writing—original draft, P.H.; Writing—review & editing, P.H.; Visualization, Y.Q.; Supervision, L.L., Y.H. and H.C.; Funding acquisition, H.C. All authors have read and agreed to the published version of the manuscript.

**Funding:** This project is supported by the National Key Research and Development Program of China (2021YFB3600701), the National Natural Science Foundation of China (62274167), the Key deployment project of the Chinese Academy of Sciences (ZDRW-XX-2022-2), the Natural Science Foundation of Zhejiang Province (LD21F040002) and the Innovation and Entrepreneurship Team of Zhejiang Province (2021R01003).

**Data Availability Statement:** Data are contained within the article and Supplementary Materials.

**Conflicts of Interest:** The authors declare no conflicts of interest.

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
