# Peer review of "Effect of In-Situ H Doping on the Electrical Properties of In2O3 Thin-Film Transistors"

_electronics, doi:10.3390/electronics13081478_

Round 1

Reviewer 1 Report

Comments and Suggestions for Authors

The in-situ H doping in In2O3 thin films and transistors is well discussed in this work. It is demonstrated that the prepared In2O3:H TFTs have sound electrical properties, including a high field effect mobility of 47.8 cm2 /Vs, a small subthreshold swing 0.25 V/dec., as well as the reduced defects and improved film quality. Before accepting this work for publication, I have several issues that are not clear to me, as listed below:

1.      In the article, the field-effect mobility in this work is much higher than the hall mobility. What’s the reason behind?

2.      In the Figure2(c), so many particles are present at the Pt region, and please explain this phenomenon.

3.      It is necessary to be polished for the written English.

For example: In the line 116,It is worth mentioning that these observation diverge from existing reports on H-doped In2O3

The ID of In2O3 TFT without H2 introduction decreased to 361 μA (VGS=1 V, VDS=15 V).” to The ID of In2O3 TFT without introducing H2 is decreased to 361 μA (VGS=1 V, VDS=15 V)

Reviewer 2 Report

Comments and Suggestions for Authors

The manuscript entitled “Effect of in-situ H doping on the electrical properties of In2O3 thin-film transistors” by Hu et. al. demonstrated the effect of H-doping in In2O3 thin film for TFT applications. The study reveals that doping contributes to diminishing surface defects within the thin film, consequently leading to a significant reduction in electron concentration. This reduction is supported by Hall effect measurements. Furthermore, TFT measurements indicate a shift in the threshold voltage (VTh) towards positive values, suggesting a decrease in electron concentration, while the decrease in subthreshold swing (SS) implies a reduction in defect states. The introduction of H doping results in notable enhancement in charge carrier mobility. Overall, the research demonstrates commendable merit and constitutes a crucial contribution to this field. However, the following questions and comments require addressing before the paper can be published.

Comments:

1.     In figure 1b, the lagend is missing. Also, the authors should include the JCPDS data of In2O3 for comparison.

2.     In figure 2a-d, authors did not discuss the (200) crystal plane which is observed for the doped samples but not for the pristine sample. Please explain this.

3.     In figure 2g, annealing in air reduced the O/In ratio significantly for the pristine sample, whereas for doped sample it increased marginally after annealing. Can authors provide a possible explanation?

Comments on the Quality of English Language

Overall English language quality of the manuscript is decent, although needs some minor corrections.

Round 2

Reviewer 2 Report

Comments and Suggestions for Authors

The authors have made the necessary amendments in the revised version which is satisfactory. I would like to recommend this work to be published in the Electronics journal.